# Rethinking Memory in Continual Learning: Beyond a Monolithic Store of the Past

**Yaqian Zhang**                                                                 *yaqianz@waikato.ac.nz*
*AI Institute, University of Waikato*

**Bernhard Pfahringer**                                                          *bernhard@waikato.ac.nz*
*AI Institute, University of Waikato*

**Eibe Frank**                                                                   *eibe@waikato.ac.nz*
*AI Institute, University of Waikato*

**Albert Bifet**                                                                 *abifet@waikato.ac.nz*
*AI Institute, University of Waikato*
*LTCI, Télécom Paris*

**Reviewed on OpenReview:** *https://openreview.net/forum?id=wgjVUIYyOD*

## Abstract

Memory is a critical component in replay-based continual learning (CL). Prior research has largely treated CL memory as a monolithic store of past data, focusing on how to select and store representative past examples. However, this perspective overlooks the higher-level memory architecture that governs the interaction between old and new data. In this work, we identify and characterize a dual-memory system that is inherently present in both online and offline CL settings. This system comprises: a short-term memory, which temporarily buffers recent data for immediate model updates, and a long-term memory, which maintains a carefully curated subset of past experiences for future replay and consolidation. We propose *memory capacity ratio* (MCR), the ratio between short-term memory and long-term memory capacities, to characterize online and offline CL. Based on this framework, we systematically investigate how MCR influences generalization, stability, and plasticity. Across diverse CL settings—class-incremental, task-incremental, and domain-incremental—and multiple data modalities (e.g., image and text classification), we observe that a smaller MCR, characteristic of *online CL*, can yield comparable or even superior performance relative to a larger one, characteristic of *offline CL*, when both are evaluated under equivalent computational and data storage budgets. This advantage holds consistently across several state-of-the-art replay strategies, such as ER, DER, and SCR. Theoretical analysis further reveals that a reduced MCR yields a better trade-off between stability and plasticity by lowering a bound on generalization error when learning from non-stationary data streams with limited memory. These findings offer new insights into the role of memory allocation in continual learning and underscore the underexplored potential of online CL approaches.[1]

## 1 Introduction

Deep neural networks are commonly trained by collecting a large dataset upfront to create independent and identically distributed (IID) mini-batches that are used to optimize network parameters using stochastic gradient descent (SGD). When learning from non-IID data where new tasks appear over time, these networks often suffer from catastrophic forgetting, where new information overwrites what has been learned previously.

---

[1]Code: https://github.com/YaqianZhang/long-term-short-term

Algorithms for continual learning (CL) aim to address this by enabling the acquisition of new skills and knowledge with minimum impact on what has been learned in the past. Two primary problem settings are considered in this context: offline CL and online CL. Offline CL assumes that when a new task needs to be learned, a new training dataset becomes available from which IID mini-batches can be drawn. In contrast, online CL operates on a continuously arriving stream of data. Instead of waiting for a complete dataset, the model begins updating as soon as mini-batches of new data arrive.

A major challenge in both online and offline CL is balancing the acquisition of new knowledge, referred to as "plasticity', with the retention of previous knowledge, known as "stability", especially when resources like memory and computation are limited. Replay-based methods, which maintain a subset of past samples, often referred to as exemplars, to adjust gradients during optimization, have shown promise in mitigating forgetting. However, previous studies that applied these methods in online CL Mai et al. (2022); Soutif-Cormerais et al. (2023) often report lower performance compared to studies that applied the same methods in offline CL Masana et al. (2022); Buzzega et al. (2020). This has led to the widespread impression that online continual learning is inherently more difficult. To better conceptualize this difficulty, recent work (Zhang et al., 2022; Jung et al., 2022) has highlighted the problem of insufficient plasticity: without the ability to store the full data distribution of new tasks and train for multiple epochs, online methods are more prone to underfitting of the new task.

In this paper, we challenge this notion by showing that the performance limitations seen in online CL are primarily due to constrained memory and computing resources rather than the online setting itself. Perhaps more interestingly, we show that, depending on the replay mechanism used, online CL can achieve greater plasticity than offline CL. For instance, we demonstrate that the online version of DER++ (Buzzega et al., 2020) exhibits greater plasticity than its offline counterpart. More specifically, we provide a controlled comparison between online and offline CL by accounting for disparities in compute and memory budgets and introduce the "memory capacity ratio" (MCR) to represent the relative sizes of short-and long-term memory. Here, short-term memory refers to the storage space available to store new data, and long-term memory represents the capacity available to store past data. We demonstrate that the fundamental distinction between online and offline CL lies in their MCRs: online CL operates with a smaller short-term memory equal to the size of incoming streaming batch size, while offline CL utilizes a larger one equal to the size of the task data (see Algorithm 1).

Since the short-term memory contains new information that the model has not encountered, and long-term memory stores previously learned information that has been repeatedly trained, the capacity ratio between these two memory types is closely related to the stability and plasticity of continual learning systems. We systematically investigate two fundamental open questions: "*How does the MCR affect stability, plasticity, and generalization in experience replay?*" and "*How does the effect of the MCR interact with forgetting mitigation strategies (e.g., contrastive replay or knowledge distillation regularization) and the characteristics of the learning problem (e.g., whether it is class incremental, domain incremental or task incremental)?*".

We conduct a theoretical analysis based on a bound on the generalization error in a simplified setting with 0-1 classification loss showing that 1) the effect of the MCR depends on task similarity and problem structure; 2) a small MCR can lead to a lower bound under certain conditions (Section 4). Subsequently, we investigate whether this theoretical result can be extended to practical CL algorithms with various complex forgetting mitigation strategies. We empirically study the effect of MCR by controlling the memory budget, training data, forgetting mitigation strategies, and training regimes. The generalization performance of continual learning is measured by the accuracy of a separate test set representing all the information to be learned. Across four state-of-the-art replay strategies, we observe that a smaller MCR leads to comparable or superior performance compared to a larger MCR value.

Contrary to conventional wisdom, we show that online CL, with a smaller MCR, does not necessarily compromise plasticity compared to offline approaches. Our analysis reveals that, depending on the replay mechanism and regularization design employed, reducing the MCR can lead to three types of changes in stability and plasticity: a) improved stability at the cost of reduced plasticity, b) enhanced plasticity with reduced stability, or c) simultaneous improvements in both stability and plasticity (Section 6). Overall, our work provides new insights into memory allocation dynamics in continual learning and highlights the potential

---

**Algorithm 1:** Unified continual replay framework instantiated with experience replay

---

    `// Non-stationary data stream:` $\mathcal{D}_t = \cup_t \mathcal{X}_t$`, where` $\mathcal{X}_t$ `is the incoming batch`
    `// Short-term memory` $\mathcal{M}_{short}$`:  storing new data`
    `// Long-term memory` $\mathcal{M}_{long}$`:  storing past data`
    `// offline CL:` $|\mathcal{M}_{short}|$ `is equal to the task size` $|\cup_{T_i \leq t < T_{i+1}} \mathcal{X}_t|$
    `// online CL:` $|\mathcal{M}_{short}|$ `is equal to the stream batch size` $|\mathcal{X}_t|$
    `// The memory management policy` $\pi$ `selects samples from` $\mathcal{M}_{short}$ `for storage in` $\mathcal{M}_{long}$`.`

**1** `function` `ContinualLearning`($\mathcal{X}_t$`,`$\theta$`,`$\mathcal{M}_{short}$`,`$\mathcal{M}_{long}$`,`)
**2**     $\mathcal{M}_{short} \leftarrow \mathcal{M}_{short} \cup \mathcal{X}_t$ `// Update short-term memory`
**3**     `if` $\mathcal{M}_{short}$ `is full` `then`
**4**         $\theta \leftarrow$ `ModelTraining`($\theta$`,`$\mathcal{M}_{short}$`,`$\mathcal{M}_{long}$)
**5**         $\mathcal{M}_{long} \subset_\pi \mathcal{M}_{short} \cup \mathcal{M}_{long}$ `// Update long-term memory`
**6**         $\mathcal{M}_{short} \leftarrow \emptyset$
**7**     `return` $\theta$`,`$\mathcal{M}_{short}$`,`$\mathcal{M}_{long}$

**8** `function` `ModelTraining`($\theta$`,`$\mathcal{M}_{short}$`,`$\mathcal{M}_{long}$)
**9**     `for` K epochs `do`
**10**         `for` $\mathcal{B}_{short}$ `in` $\mathcal{M}_{short}$ `do`
**11**             sample $\mathcal{B}_{long}$ from $M_{long}$,
**12**             $\theta \leftarrow \theta - \eta \nabla L(\mathcal{B}_{short} \cup \mathcal{B}_{long}; \theta)$ `// Using basic experience replay as an example`
**13**     `return` $\theta$

---

of online CL to achieve superior performance particularly in environments where memory and computational resources are constrained.

## 2 Related Work

**Continual Learning**. General continual learning (Delange et al., 2021; Buzzega et al., 2020) is an idealized scheme for learning from an infinite data stream, with desiderata like constant memory, online learning, no task boundaries, no task labels, and graceful forgetting. Various relaxations exist with different assumptions. Early work focused on so-called "task-incremental" settings (Mallya & Lazebnik, 2018; Serra et al., 2018) that assume access to task labels during training/testing. Despite promising results, relying on a task oracle is impractical. Recent so-called "class-incremental" and "domain-incremental" learning approaches remove this assumption (Mirza et al., 2022; Masana et al., 2022; Van de Ven & Tolias, 2019). Nevertheless, these methods still require the knowledge of task boundaries to allow multi-epoch training over tasks. In contrast, online continual learning (Chaudhry et al., 2019; Aljundi et al., 2019; Mai et al., 2022) does not collect a specific task dataset but rather trains the model incrementally as mini-batches of new data become available.

**Memory and Compute**. In the literature, online and offline continual learning are studied separately with different memory and computational setups (see Table B in the Appendix B). Studies and surveys of offline continual learning usually utilize 50-250 training epochs per task on standard benchmarks. Online continual learning, by design, can only use a single "epoch", but the number of gradient update steps can be increased through repeated iterations yielding multiple gradient-based updates for each incoming batch (Zhang et al., 2022; Soutif-Cormerais et al., 2023). However, it is common practice to limit the number of iterations to fewer than 10, potentially putting online learning at a disadvantage. Considering storage costs, a common practice in continual learning is to compare different online or offline methods based on a fixed budget for storing exemplars, a subset of past samples. This approach views memory in continual learning as a single system comprised solely of past exemplars. This perspective overlooks the higher-level memory architecture that governs the interaction between old and new data in dealing with a continuous data stream. In contrast, we consider the actual amount of data that needs to be stored, comprising both short-term and long-term memory. By controlling the total storage required to hold data, referred to as memory, and the number of

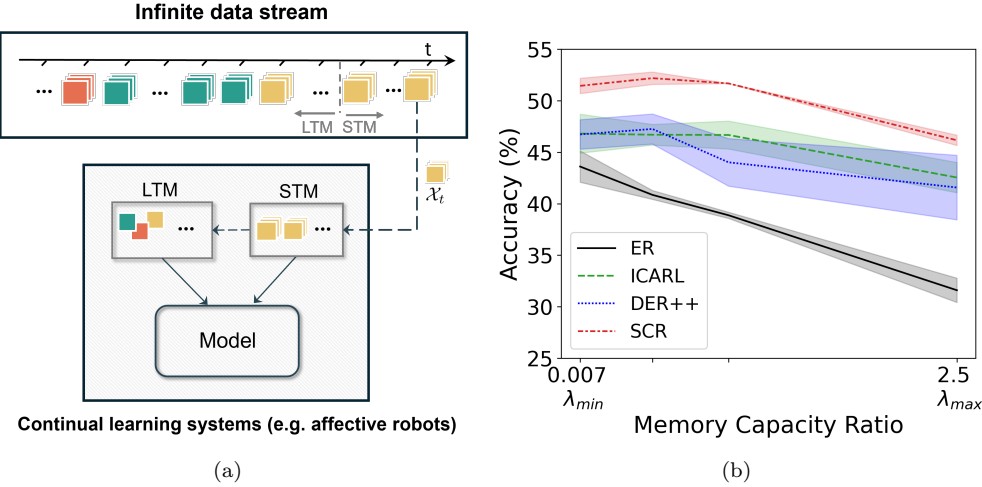

Figure 1: Long-term and short-term memory structure in continual replay: (a) The short-term memory (STM) greedily stores recent incoming data and the long-term memory (LTM) selectively stores a subset of data after model training. (b) The effect of the memory capacity ratio (MCR=STM/LTM): reducing MCR significantly boosts the learning efficacy of continual learning algorithms including ER, iCARL, DER++, SCR on Split-Mini-ImageNet.

iterations yielding gradients, we examine the effect of relative capacity of long-term and short-term memory on learning efficacy.

**Theoretical Study in Continual Learning**. There are several theoretical studies deriving bounds on generalization performance for continual learning. (Peng et al., 2023; Doan et al., 2021) provide theoretical analysis with PAC-Bayes bounds. Meanwhile, Ye & Bors (2022) derives a generalization bound for the 0–1 loss using discrepancy distance and Rademacher complexity. Although closely related to our work, their analysis considers only the effect of replayed exemplars and does not account for the contribution of incoming data that is simultaneously used for model updates. In contrast, we address this limitation by explicitly modeling the influence of samples stored in long-term memory as well as those held in short-term memory in our generalization bound.

# 3 Rethinking CL Memory: Beyond a Monolithic Store of Past Data

## 3.1 Problem Setup

We consider the general continual learning problem proposed in Delange et al. (2021), which deals with infinite non-stationary data streams, and approach this problem with a unified continual replay framework.

**Non-stationary Data Stream**. At each time step $t$, a continual learning algorithm $\mathcal{A}$ receives an incoming batch of data samples of size $B$, $\mathcal{X}_t = \{\mathbf{x}_i, y_i\}_{i=1,..,B}$, drawn from the current data distribution $\mathbb{P}_t$. Each batch forms part of a non-stationary (potentially infinite) stream of data $\mathcal{D}_t = \cup_t \mathcal{X}_t$. The distribution $\mathbb{P}_t$ may change at any time. A basic approach to continual learning from such a stream of data would be to attempt to minimize the empirical risk on all the data seen so far:

$$\min_{\theta} \mathcal{R}_t(\theta) = \min_{\theta} L(\cup_t \mathcal{X}_t; \theta). \tag{1}$$

with a loss function $L$, a CL network function $f : x \to y$, and its associated parameters $\theta$.

*Task Boundaries.* The period where the data distribution $\mathcal{P}_t$ stays the same is often called a *task*. Task boundaries are defined as the time steps where changes in the data distribution occur, i.e., $\{T_i\} \doteq [t|\mathbb{P}_t \neq \mathbb{P}_{t-1}]$. Given the task boundaries of a data stream, each task data can be denoted as $\mathcal{C}_i = \cup_{T_i \leq t < T_{i+1}} \mathcal{X}_t$ and $\cup_i \mathcal{C}_i = \mathcal{D}_t$.

### 3.2 Unified Replay Framework with a Dual-Memory Structure

Prior work in continual learning typically treats memory as a monolithic buffer of past data and focuses primarily on strategies for managing stored exemplars. In this paper, we argue that there is a higher-level memory structure governing not only how new and historical data interact, but also how buffered data systems interface with a continuous data stream. This memory structure involves storage policies for both recent and past samples. We show that such a memory structure arises naturally in existing online and offline continual learning setups.

Concretely, we model this higher-level structure as a dual-memory system consisting of a short-term memory and a long-term memory, as illustrated in Figure 1. We formalize the replay process under this dual-memory view through a Unified Continual Replay (UCR) framework.

**Unified Continual Replay Framework** $UCR(M_s, M)$. Given a total data storage budget $M$ ($M < |\mathcal{D}_t|$) and a stream of data $\mathcal{D}_t = \cup_t \mathcal{X}_t$ with batch size $B$, the storage space is divided into two parts: a short-term memory of size $M_{short}^t$ ($M_s \geq B$) and a long-term memory of size $M - M_{short}$. The short-term memory $\mathcal{M}_{short}^t$ greedily stores recent batches from the data stream until it is full: $\mathcal{M}_{short}^t = \cup_{t-n+1,\dots,t} \mathcal{X}_t, n = \frac{M_{short}}{B}$. A model training session is triggered when $\mathcal{M}_{short}^t$ is full. After the model training session, $\mathcal{M}_{short}^t$ is emptied, with some of the short-term memory samples moved into $\mathcal{M}_{long}^t$ based on some sample selection policy $\pi$:
$$\mathcal{M}_{long}^{t+1} \subset_\pi \mathcal{M}_{short}^t \cup \mathcal{M}_{long}^t.$$

This framework is agnostic to the specific replay strategy. Algorithm 1 provides an example instantiation using experience replay with $K$ training epochs. Algorithm 2 in the appendix gives an example instantiation using DER++.

Crucially, online and offline CL can be considered as two extreme cases of this framework. Offline CL stores the whole task data in short-term memory based on the knowledge of task boundaries: $M_{short} = |\cup_{T_i \leq t < T_{i+1}} \mathcal{X}_t| = C$. Online CL only stores the current batch in the short-term memory: $M_{short} = |\mathcal{X}_t| = B$.

**Memory Capacity Ratio**. We define the Memory Capacity Ratio as $\lambda = M_{short}/(M - M_{short})$. Offline CL yields a large value of MCR, i.e., $\lambda_{offline} = \lambda_{max} = C/(M - C)$, where $C$ is the task size, while online CL yields a small MCR: $\lambda_{online} = \lambda_{min} = B/(M - B)$, where $B$ is the streaming batch size.

*Semi-offline CL.* Apart from the two extreme cases, representing offline and online continual learning respectively, the unified framework enables investigation of a novel semi-offline setting, where the short-term memory stores more than one streaming batch, and $B < M_{small} < C$. We investigate what the optimal value for $\lambda$ is in $[\lambda_{min}, \lambda_{max}]$.

## 4 Theoretical Study on Generalization Bound with 0-1 Loss

To theoretically understand the effect of MCR on continual learning efficacy, we consider a bound on generalization error for 0-1 loss. To analyze generalization in a non-IID setting, we follow work on domain adaptation (Mansour et al., 2009) with 0-1 loss and derive a bound based on the concept of *discrepancy distance*.

**Definition 1** (Discrepancy Distance). Mansour et al. (2009). Let $H$ be a set of functions mapping $X$ to $Y$ and let $L : Y \times Y \to R^+$ define a loss function over $Y$. The discrepancy distance between two distributions $Q_1$ and $Q_2$ over $X$ is defined by

$$\text{disc}_L (Q_1, Q_2) \doteq \max_{h,h' \in H} |\mathcal{L}_{Q_1} (h', h) - \mathcal{L}_{Q_2} (h', h)|.$$

where the expected loss of two functions over a distribution is denoted as $\mathcal{L}_{\mathbb{Q}}(f, g) \doteq \mathrm{E}_{x \sim \mathbb{Q}}[L(f(x), g(x))]$.

We consider the stored data (i.e., data stored in the long-term and short-term memory) as the source domain, and the test data of the data stream as the target domain. Let $\mathbb{D}_t$ and $\mathbb{M}_t$ denote the true probability distributions of the data stream and the stored samples at time step $t$ respectively. Let $\hat{\mathbb{M}}_t$ denote the empirical distribution of stored samples with a finite sample size of $M$. The true labeling function of all

the data seen so far is defined as $h_y^t$ [2]. Given the optimal solutions $h_{\mathbb{M}_t}^* \doteq \arg\min_{h \in H} \mathcal{L}_{\mathbb{M}_t}\left(h, h_y^t\right)$ and $h_{\mathbb{D}_t}^* \doteq \arg\min_{h \in H} \mathcal{L}_{\mathbb{D}}\left(h, h_y^t\right)$, the generalization bound is presented in Theorem 1[3].

**Theorem 1**. Let $H$ be a hypothesis set bounded by some $A_0 > 0$ for the loss function $L : L(h, h') \leq A_0$. For all $h, h' \in H$, assume that the loss function $L$ is symmetric and obeys the triangle inequality. Then, for any $h \in H$ and any $\delta > 0$, with probability at least $1 - \delta$, the following generalization bound holds:

$$\mathcal{L}_{\mathbb{D}_t}(h, h_y^t) \leq \mathcal{L}_{\hat{\mathbb{M}}_t}(h, h_{\mathbb{M}_t}^*) + \widehat{\Re}_{\mathcal{M}_t}(H) + 3A_0\sqrt{\frac{\log\frac{2}{\delta}}{2M}} + \mathrm{disc}_L(\mathbb{D}_t, \mathbb{M}_t) + \mathcal{L}_{\mathbb{D}_t}(h_{\mathbb{M}_t}^*, h_{\mathbb{D}_t}^*) + \mathcal{L}_{\mathbb{D}_t}(h_{\mathbb{D}_t}^*, h_y^t), \quad (2)$$

where $\widehat{\Re}_{\mathcal{M}}(H)$ is the empirical Rademacher complexity of the hypothesis set over the stored samples $\mathcal{M}_t = \mathcal{M}_{short}^t \cup \mathcal{M}_{long}^t$.

The proof follows derivations of Theorem 8 in Mansour et al. (2009) and Theorem 1 in Ye & Bors (2022) and is presented in Appendix A. Given the high expressivity of deep networks, $\mathcal{L}_{\mathbb{D}_t}(h_{\mathbb{D}_t}^*, h_y^t)$ approaches zero.

The memory management policy governs the relationship between the memory distribution ($\mathbb{M}_t$) and the data stream ($\mathbb{D}_t$). While Theorem 1 holds regardless of their similarity, the alignment between the two affects the tightness of the bound through the term $\mathcal{L}_{\mathbb{D}_t}(h_{\mathbb{M}_t}^*, h_{\mathbb{D}_t}^*)$ on the right hand side of Theorem 1. Since $\mathbb{D}_t$ and $\mathbb{M}_t$ both cover the new and past task domains, we assume $\mathcal{L}_{\mathbb{D}_t}(h_{\mathbb{M}_t}^*, h_{\mathbb{D}_t}^*)$ to be reasonably small and thus the bound is primarily driven by $\mathrm{disc}_L(\mathbb{D}_t, \mathbb{M}_t)$.

**Remarks**. Theorem 1 highlights that a key factor influencing generalization performance is the discrepancy $\mathrm{disc}_L(\mathbb{D}_t, \mathbb{M}_t)$, which measures the distribution difference between the stored data and the data stream. While much research focuses on constructing representative exemplars (the long-term memory), the role of the incoming buffer (short-term memory) is often overlooked (Peng et al., 2023; Ye & Bors, 2022). Theorem 1 reveals that it is the joint distribution of the long-term and short-term memories, rather than the long-term memory alone, that determines the generalization bound.

We analyze the discrepancy distance $\mathrm{disc}_L(\mathbb{D}_t, \mathbb{M}_t)$ at task boundaries in Proposition 1.

**Proposition 1.** Assume $\mathbb{D}_t^+$ denotes the probability distribution of the most recent task $\mathcal{C}_i$ and $\mathbb{D}_t^-$ denotes the probability distribution of all past tasks seen so far $\cup_{1,\ldots,i-1}\mathcal{C}$. Given the number of samples seen in the data stream $N_t = \sum_t |\mathcal{X}_t|$ and the number of samples seen in the previous tasks $N_t^- = \sum_{k=1}^{k=i-1} |\mathcal{C}_k|$, we have:

$$\mathrm{disc}_L(\mathbb{D}_t, \mathbb{M}_t) = \frac{N_t^- \lambda(N_t - M)}{N_t(N_t + \lambda N_t - \lambda M)} \mathrm{disc}_L(\mathbb{D}_t^-, \mathbb{D}_t^+). \quad (3)$$

The proof is based on the property of reservoir sampling Vitter (1985) and is shown in Appendix A.2. Proposition 1 reveals the effect of $\lambda$, problem structure, and task similarity, on the discrepancy distance.

Based on Eq 9, we quantitatively compute the effect of MCR as:

**Corollary 1**.
$$\nabla_\lambda \mathrm{disc}_L = \frac{N_t^-(N_t - M)}{(N_t + \lambda N_t - \lambda M)^2} \mathrm{disc}_L(\mathbb{D}_t^-, \mathbb{D}_t^+) \quad (4)$$

Equation 4 reveals several insights:

1. The effect of the MCR $\lambda$: When $N_t > M$ and $\mathrm{disc}_L(\mathbb{P}_t^-, \mathbb{P}_t^+) \neq 0$, we have $\nabla_\lambda \mathrm{disc}_L > 0$. This suggests a smaller $\lambda$ leads to a lower discrepancy distance and a lower generalization bound.

2. Bounded vs. unbounded memory: In the case of $N_t = M$, i.e., an unbounded memory system for learning data streams, we have $\nabla_\lambda \mathrm{disc}_L = 0$. In this case, the choice of MCR does not affect the generalization bound.

---

[2]Throughout this work, we assume the true labeling function $h_y^t$ is deterministic Ben-David & Urner (2014), i.e., $y = h_y^t(x)$ uniquely maps each input $x$ to a single label $y$. This excludes cases with label noise (Frénay & Verleysen, 2013), inherent ambiguity (e.g., subjective annotations), or stochastic data-generating processes. Extensions to probabilistic labeling functions $P(y|x)$ are left for future work.

[3]Although this analysis is focused on classification tasks, a similar derivation can be performed for regression tasks with MSE loss.

A related result is $\frac{\partial^2 \operatorname{disc}_L}{\partial N \partial \lambda} > 0$ and $\frac{\partial^2 \operatorname{disc}_L}{\partial M \partial \lambda} < 0$. This suggests the effect of MCR becomes more evident with a longer data stream $N_t$ and a more limited memory budget $M$.

3. Task similarity: The effect of MCR also depends on task similarity $\mathbb{P}_t^-$ and $\mathbb{P}_t^+$, and the loss function $L$. With smaller $\operatorname{disc}_L(\mathbb{P}_t^-, \mathbb{P}_t^+)$, the effect of MCR diminishes.

# 5 Empirical Study on Forgetting Mitigation Strategies and Training Dynamics

While our previous analysis of MCR's effect on generalization bounds reveals important theoretical insights about the relationships between MCR, task similarity, and problem structure, several key limitations remain unaddressed.

**Forgetting Mitigation Losses.** The generalization bound analysis assumes symmetric loss functions. However, modern continual learning methods typically employ complex asymmetric loss functions (e.g., cross-entropy and contrastive losses) with some auxiliary loss designs, like regularization loss terms based on knowledge distillation. It is unclear how these forgetting mitigation strategies interplay with the long-term and short-term allocation mechanisms. Intuitively, the incorporation of forgetting mitigation loss may reduce the necessary size of long-term memory.

**Training Dynamics.** In addition, the theoretical framework of generalization bounds primarily examines relationships between model complexity, sample size, and generalization error, without explicitly considering training dynamics, such as the effects of SGD optimization and model initialization. This limitation is especially relevant to continual learning, where model initialization plays a fundamental role: at the start of each new task, the model has already been optimized for previous tasks. This creates an initial state where the model performs well on data stored in long-term memory but poorly on new data in short-term memory. This performance disparity at initialization may also reduce the necessary size of long-term memory and thus influence the effect of the MCR.

To address these limitations, we investigate the effect of the MCR empirically, by considering practical CL training dynamics and forgetting mitigation strategies. We investigate whether the previous theoretical results hold with complex loss functions and training dynamics.

**Control Variables**. To isolate the effect of the MCR, we consider the following control factors in our empirical study:

- *Stored Data Size (M)*: We compare the effect of the MCR under a fixed total memory budget $M$. This ensures the training data size used in each model update session is consistent and eliminates potential confounding factors related to the training data size.

- *Seen Data Size (N)*: We assume that all samples of the incoming batch will be fitted in the short-term memory, i.e., $M_{short} \geq B$. This guarantees that all samples in the data stream will be used by the model at some stage, regardless of MCR value.

- *Compute Resources (K)*: We control the number of iterations to be the same when comparing the effect of the MCR. This ensures a fair comparison of the computational resources utilized across different experimental conditions.

- *Algorithms and Hyperparameters*: We compare the effect of the MCR under the exact same CL algorithms, using the same data augmentation techniques and hyperparameter settings. This allows for an isolated evaluation of the impact of the MCR.

## 5.1 Experiment Setup

**Image Classification**. The main experiments involve three standard CL benchmarks based on image classification: Split-CIFAR100 with 20 tasks, Split-Mini-ImageNet Vinyals et al. (2016) with 10 tasks, and CORE50 Lomonaco & Maltoni (2017) with 9 tasks. Table 1 lists the image size, the number of classes, the number of tasks, and data size per task for the three CL benchmarks. The task size in these three benchmarks

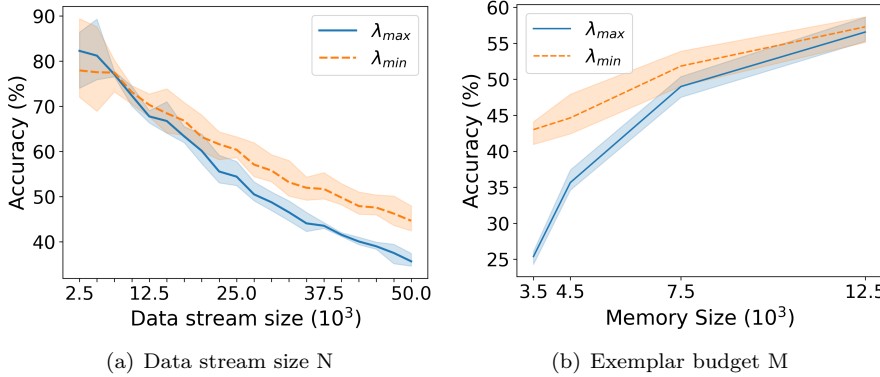

(a) Data stream size N          (b) Exemplar budget M

Figure 2: Memory Capacity Ratio and problem structure: the effect of MCR ($\lambda$) on continual learning efficacy becomes more evident with a longer data stream sequence $N$, and a smaller exemplar budget $M$. This experiment employs ER in CIFAR100 with 20 tasks.

is 2500, 5000, and 12000 respectively. We consider the data stream batch size to be 50. We use ResNet-18 for all experiments. We apply standard data augmentation (random cropping and flipping) in most experiments. For SCR and CORE50, we additionally use color jittering and random grayscale conversion. Hyperparameter details can be found in the Appendix D.

Table 1: Dataset statistics for continual image classification.

|  | Image Size | #Task | # Class | Train per task | Test per task |
|---|---|---|---|---|---|
| Split-CIFAR100 | 3x32x32 | 20 | 100 | 2,500 | 250 |
| Split-Mini-ImageNet | 3x84x84 | 10 | 100 | 5,000 | 1,000 |
| Split-CORE50-NC | 3x128x128 | 9 | 50 | 12,000 | 4,500 |

**Text Classification**. We also consider continual text classification with three text classification datasets, including AG News (news classification), Yelp (sentiment analysis), and Yahoo! Answer (Q&A classification). A summary of the text datasets is shown in Table 2.

Following (de Masson D'Autume et al., 2019; Huang et al., 2021), we utilize the pretrained BERT-base-uncased model from the HuggingFace Transformers Wolf et al. (2020) library as the base feature extractor. The experiments are conducted with a batch size of 8. The learning rate is set to 3e-5 and the weight decay for all parameters is set to 0.01.

**Evaluation Metric**. The performance of CL is measured by the final accuracy after training on all tasks, defined as $A_T = \frac{1}{T} \sum_{j=1}^{j=T} a_{T,j}$, where $a_{i,j}$ denotes the model's accuracy on the held-out test set of task $j$ after training on task $i$. We conduct all experiments across three random seeds to account for potential variability factors, including task splitting in continual learning benchmarks, random sample selection in reservoir sampling, random model initialization and stochastic gradient descent. The plots are presented with 95% confidence intervals to provide a measure of statistical significance. Furthermore, the tables report the mean values along with the corresponding standard deviations.

Table 2: Dataset statistics for continual text classification.

| Order | Dataset | Type | #Class | Train | Test |
|---|---|---|---|---|---|
| 1 | AGNews | News | 4 | 8000 | 7600 |
| 2 | Yelp | Sentiment | 5 | 8000 | 7600 |
| 3 | Yahoo | Q&A | 10 | 20000 | 7600 |

Table 3: The effect of memory capacity ratio in different benchmarks and algorithms. Given a memory budget $M$, and a data stream with task size $C$ and batch size $B$, small MCR $\lambda_{min} = B/(M - B)$ values lead to a significant performance advantage over large MCR values $\lambda_{max} = C/(M - C)$.

| | MCR | ER | iCARL | DER++ | SCR |
|---|---|---|---|---|---|
| S-CIFAR100-20 | $\lambda_{max}$ | $35.1 \pm 0.8$ | $44.8 \pm 1.1$ | $47.2 \pm 0.4$ | $45.1 \pm 0.4$ |
| | $\lambda_{min}$ | $44.6 \pm 2.4$ | $50.0 \pm 1.6$ | $50.9 \pm 0.9$ | $51.9 \pm 0.5$ |
| | | $9.5 \uparrow$ | $5.2 \uparrow$ | $3.7\uparrow$ | $6.8 \uparrow$ |
| S-MINI-IMAGENET-10 | $\lambda_{max}$ | $31.2 \pm 1.1$ | $42.5 \pm 1.2$ | $41.0 \pm 2.7$ | $46.3 \pm 0.4$ |
| | $\lambda_{min}$ | $43.7 \pm 1.2$ | $46.8 \pm 1.5$ | $46.4 \pm 1.3$ | $51.8 \pm 0.7$ |
| | | $12.5 \uparrow$ | $3.3 \uparrow$ | $5.4 \uparrow$ | $5.5\uparrow$ |
| S-CORE-9 | $\lambda_{max}$ | $40.1 \pm 2.4$ | $45.8 \pm 1.6$ | $38.2 \pm 1.5$ | $62.1 \pm 1.3$ |
| | $\lambda_{min}$ | $50.7 \pm 1.7$ | $50.1 \pm 1.6$ | $46.7 \pm 2.6$ | $69.7 \pm 0.3$ |
| | | $10.6 \uparrow$ | $4.3 \uparrow$ | $8.5 \uparrow$ | $7.6$ |

**Replay Strategies**. We consider three types of replay-based approaches: 1) *Direct rehearsal.* Our main experiment is focused on experience replay Chaudhry et al. (2019), which is a simple approach that achieves competitive performance especially in large-scale settings Prabhu et al. (2023). ER incorporates past exemplars directly in the training via cross-entropy loss. 2) *Knowledge distillation.* Many replay-based methods leverage knowledge distillation to construct a regularization loss, using a past model as the teacher and the current model as the student (Li & Hoiem, 2017; Rebuffi et al., 2017; Buzzega et al., 2020; Hinton et al., 2015). A classic method is iCaRL which maintains a past model and computes distillation loss based on the past network's outputs related to old classes. Instead of computing logits based on a past model, the distillation loss of DER++ uses the network's logits sampled throughout the optimization trajectory, and the distillation loss is computed over past exemplars. 3) *Contrastive replay.* Some recent works (Cha et al., 2021; Mai et al., 2021; Khosla et al., 2020) investigate the use of self-supervised learning techniques to learn a strong representation to reduce forgetting. A representative method is SCR Mai et al. (2021), which replaces cross-entropy with contrastive loss to capture more information about exemplars and achieves state-of-the-art performance.

**Memory and Compute**. To make the experiment results comparable to existing work, we follow the standard practice in the offline continual learning literature. We set the total memory budget to be task size plus 2000. More specifically, the total memory budget is 4500, 7000 and 14000 for CIFAR100, Mini-Imagenet and CORE50 respectively. With this setup, the results obtained with $\lambda_{max}$ are aligned with the existing offline continual learning results that employ a 2000 exemplar budget. Regarding the computational resources, we employ 50 training epochs ($K = 50$) as the default setting.

## 5.2 Main Findings

**Forgetting-Mitigation Strategies**. Different replay techniques make use of exemplars to preserve past knowledge in different ways. Intuitively, a replay method with a very strong forgetting mitigation design may not need a large number of exemplars from previous tasks to perform well. Thus, one interesting question is how the effect of the MCR changes with different forgetting designs. Fig 1 presents the results of different CL strategies on Split-Mini-ImageNet. Interestingly, our results show that a small MCR $\lambda$ seems to lead to better performance across different algorithms including ER, iCaRL, DER++, and SCR. The performance boost in direct rehearsal ER seems larger than in other replay methods. Table 3 presents detailed results comparing the results of the smallest MCR $\lambda_{min}$ and the largest $\lambda_{max}$ on three standard CL benchmarks. $\lambda_{min}$ leads to consistent performance improvement over $\lambda_{min}$ across three datasets (ER: $10 - 12\%$, iCaRL: $4 - 6\%$, DER++: $3 - 8\%$, SCR: $5 - 7\%$).

**Continual Learning Problem Structure**. We further investigate whether the effect of MCR is influenced by different problem structures along two dimensions: 1) *Data stream size* - Fig 2 a) suggests that with longer task sequences the advantage of small MCR becomes more obvious. 2) *Memory budget* - Fig 2 b) presents results of different sizes of memory budget, including 3500, 4500, 7500, and 12500. Given the limited memory

Table 4: The effect of memory capacity ratio $\lambda$ in task-incremental, domain-incremental and pretrained class-incremental settings.

| CL Settings | Task-Incremental | Domain-Incremental | Pretrained ResNet | Pretrained Transformer |
|---|---|---|---|---|
| Dataset | CIFAR100 | CLRS | Mini-ImageNet | Text classification |
| $\lambda_{max}$ | **83.4 ± 1.3** | 34.2 ± 2.6 | 36.6 ± 0.9 | 65.8 ± 0.5 |
| $\lambda_{min}$ | 83.1 ± 2.3 | **36.8 ± 1.3** | **48.3 ± 1.1** | **71.9 ± 0.3** |

Table 5: The effect of MCR on different sample selection methods, including MIR and ASER.

| Method | MCR | Mini-ImageNet | | CIFAR-100 | |
|---|---|---|---|---|---|
| | | $M_{short}$ | Accuracy | $M_{short}$ | Accuracy |
| MIR | $\lambda_{min}$ | 50 | **43.4 ± 0.3** | 50 | **43.4 ± 3.1** |
| | $\lambda_{mid}$ | 2500 | 40.6 ± 0.5 | 1250 | 41.4 ± 1.4 |
| | $\lambda_{max}$ | 5000 | 30.8 ± 0.6 | 2500 | 33.4 ± 2.8 |
| ASER | $\lambda_{min}$ | 50 | **40.6 ± 0.8** | 50 | **42.8 ± 1.7** |
| | $\lambda_{mid}$ | 2500 | 38.2 ± 0.6 | 1250 | 41.0 ± 2.2 |
| | $\lambda_{max}$ | 5000 | 31.1 ± 1.0 | 2500 | 33.8 ± 1.2 |

budget, the advantage of a small MCR is more evident. When the budget is larger than 12500 (around 5 tasks), the performance of replay remains the same regardless of the choice of MCR.

**Different Continual Learning Settings.** So far, our experiments focused on the class-incremental setting and training the model from scratch. We now investigate this phenomenon in other settings. The effect of the MCR $\lambda$ in task-incremental (TI), domain-incremental (DI) and pretrained class-incremental settings is shown in Table 4. When initializing the model with a pre-trained ResNet18, we observe a similar trend: ER with a small MCR ($48.3 \pm 1.1\%$) significantly outperforms ER with large MCR values ($36.6 \pm 0.9\%$). Additionally to what is observed with ResNet and image data, we also observe similar results in text classification experiments with pre-trained Transformer models ($\lambda_{min} : 71.9 \pm 0.3\%$ and $\lambda_{max} : 65.8 \pm 0.3\%$). However, in task-incremental learning, we observe the effect of the MCR ratio is not significant ($\lambda_{min} : 83.1 \pm 2.3\%$ and $\lambda_{max} : 83.4 \pm 1.3\%$).

**Memory Management Policies**. Table 5 presents results using different memory update and retrieval strategies, including MIR (Aljundi et al., 2019) and ASER (Shim et al., 2021). Unlike random selection, MIR retrieves samples that are most interfered—i.e., those whose predictions would be most negatively affected by upcoming parameter updates. ASER, on the other hand, selects samples based on the Adversarial Shapley value, which scores memory samples by their contribution to preserving latent decision boundaries of previously observed classes. Consistent with our main findings, Table 5 shows that reducing MCR also improves performance when using advanced sample selection methods such as MIR and ASER.

In summary, our empirical study reveals that the relative capacity of long-term and short-term memory exhibits a complex interplay with forgetting-mitigation strategies and problem structure. For certain problems, such as task-incremental learning or scenarios with a large memory budget, the choice of MCR does not significantly impact performance. However, in other settings like class-incremental learning with a limited memory budget, the performance of replay methods can be significantly improved by adopting smaller MCR values. To understand how MCR influences the learning efficacy of continual learning systems, we examine the stability and plasticity dynamics next.

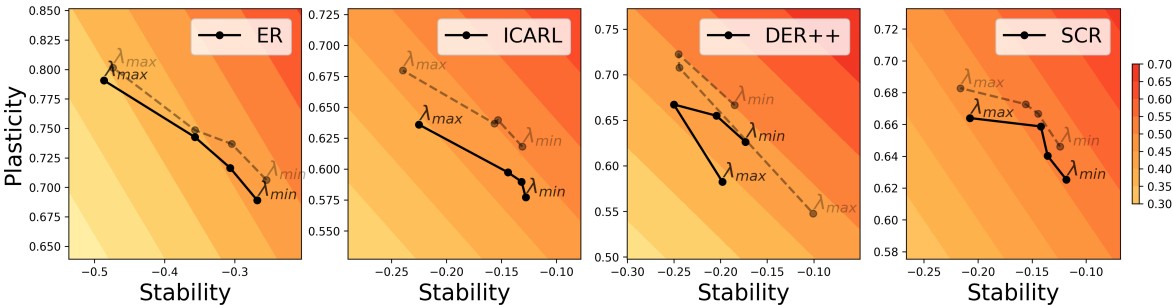

Figure 3: Stability, plasticity, and generalization across two runs with different initializations and task splits. Lower MCR yields diverse stability–plasticity patterns but consistently improves the overall stability–plasticity trade-off.

## 6   Stability and Plasticity

**Measuring Stability and Plasticity**. Beyond the evaluation of the overall performance as discussed in Section 5.1, we also examine the stability and plasticity during the learning process. Plasticity refers to the ability to learn the new tasks and is usually measured as the accuracy on the new task. Stability refers to the ability to maintain previous knowledge. There are two measures proposed in the literature related to stability: one metric is "forgetting" (Chaudhry et al., 2018), which is defined as $F_T = -\frac{1}{T-1} \sum_{i=1}^{T-1} \left(a_{T,i} - \max_{l \in 1 \dots T-1} a_{l,i}\right)$ and the related metric "backward transfer" (Lopez-Paz & Ranzato, 2017): $B_T = \frac{1}{T-1} \sum_{i=1}^{T-1} a_{T,i} - a_{i,i}$. Since the continual learning systems require a tradeoff of stability and plasticity, Zhang et al. (2022) decompose the overall CL performance into a stability-based term and a plasticity-based term, as follows:

$$A_T = \underbrace{\frac{1}{T}\Sigma_{i=1}^{T} a_{i,i}}_{\text{Plasticity}} + \underbrace{\frac{T-1}{T} B_T}_{\text{Stability}} \geq \frac{1}{T}\Sigma_{i=1}^{T} a_{i,i} - \frac{T-1}{T} F_T.$$

**Insights into Stability-Plasticity Dynamics**. Using the definitions of stability, plasticity, and generalization from Zhang et al. (2022), we plot the effect of the MCR on these metrics in Fig 3. Since a lower MCR allocates more space for storing past exemplars and less space for new samples from the current task, an intuitive expectation would be that reducing the MCR leads to better stability and worse plasticity. Surprisingly, this behavior is observed only in some algorithms (ER, iCaRL, and SCR). For other algorithms (e.g., DER++), reducing MCR leads to various types of stability-plasticity behaviors, including better plasticity and better stability at some stages, and better plasticity and worse stability at other stages, as shown in Fig. 3. These behaviors in DER++ suggest that increasing the ratio of long-term memory can aid in the learning of new tasks at certain stages.

To understand why increasing long-term memory can contribute to plasticity in some cases, we consider the distribution composition of long-term buffers. The long-term memory contains a subset of past data, and its coverage is determined by where the data stream is split. As illustrated in Fig 1, the data stream is split into two parts based on the size of the short-term memory ($M_{short}$): one part contains the recent $M_{short}$ samples, which are greedily stored in the short-term memory, and the other part contains all the samples before $M_{short}$, which are selectively captured by the long-term memory. With reservoir sampling, the long-term memory stores a fraction $p_{new}$ of new task exemplars and $1 - p_{new}$ of old task exemplars: $p_{new} = \frac{C - M_{short}}{N - M_{short}}$, where $N$ is the data stream size and $C$ is the task size. When $M_{short} < C$ (i.e., in online and semi-offline CL), we have $p_{new} > 0$, meaning the long-term memory contributes to learning the current task, promoting plasticity. We take DER as an example to illustrate how the MCR may lead to complex stability-plasticity changes. The analysis of other algorithms follows a similar manner and can be found in Appendix C. We rewrite the gradient of DER as follows:

$$\begin{aligned}
\nabla \mathcal{L}_{DER} &= \mathbb{E}_{\mathcal{M}_{short}}[\nabla \ell_{ce}] + \mathbb{E}_{\mathcal{M}_{long}}[\nabla \ell_{mse}^{kd}] \\
&= \mathbb{E}_{\mathcal{M}_{short}}[\nabla \ell_{ce}] + p_{new} \times \mathbb{E}_{\mathcal{M}_{long}^{new}}[\nabla \ell_{mse}^{kd}] + (1 - p_{new}) \times \mathbb{E}_{\mathcal{M}_{long}^{old}}[\nabla \ell_{mse}^{kd}]
\end{aligned} \tag{5}$$

As shown in Eq 5, when the ratio of short-term memory is reduced, the first term of the right-hand side (RHS) suffers from greater overfitting of new task data, which harms plasticity. However, the second term of the RHS drives plasticity, and its contribution increases with smaller $M_{short}$ because $\nabla_{M_{short}} p_{new} < 0$. Therefore, the overall plasticity is determined by the interplay between the first and second terms. DER employs different loss functions for the first term (cross-entropy) and second term (logits-based knowledge distillation), thus creating the complex plasticity dynamics in Fig 3.

## 7  Discussion and Conclusion

A key challenge in continual learning is optimizing memory usage to enhance learning efficiency. While many prior works have investigated different strategies for selecting and storing representative past data, the higher-level memory architecture governing the interaction between old and new data remains understudied. In this work, we identify and formalize a unified dual-memory system that naturally arises in both online and offline CL settings. This system comprises: (1) short-term memory, which temporarily buffers recent data for immediate model updates, and (2) long-term memory, which maintains a carefully curated subset of past experiences for future replay and consolidation. We investigate how the memory capacity ratio (MCR), defined as the relative allocation of resources between STM and LTM, affects learning efficacy in non-stationary streams. Empirically, we find that reducing the MCR induces distinct stability-plasticity trade-offs depending on the replay mechanism: some replay configurations improve both stability and plasticity, others enhance one at the expense of the other, yet a smaller MCR consistently matches or outperforms larger MCRs in generalization. Theoretical analysis of generalization bounds in a simplified setting further supports this result.

We believe the impact of MCR has likely remained unexplored in prior work because 1) CL memory is often treated as a monolithic store of exemplars, and 2) online/offline CL are typically studied in isolation with divergent setups. By re-examining continual replay through the lens of short- and long-term memory, our work offers a more comprehensive understanding of replay dynamics in non-stationary environments. Our findings suggest that designing a memory structure with a small MCR can be advantageous in building practical continual learning systems, such as those used in affective robotics (Churamani et al., 2020), smart home assistants, or wearable devices.

**Limitations.** This work examines rehearsal and knowledge distillation techniques for continual learning. Other approaches, such as correcting task recency bias (Wu et al., 2019; Hou et al., 2019) or expanding network capacity (Zhou et al., 2022; Yan et al., 2021), are not covered by our analysis. Additionally, we use a fixed MCR throughout the continual learning process. Since MCR directly influences the trade-off between stability and plasticity, dynamically adjusting MCR to achieve a better balance between the two represents a promising direction for future research. Moreover, with a time-varying MCR, an interesting avenue for exploration is designing an adaptive memory management strategy that aligns with the changing memory size while maintaining representativeness across different tasks.

## 8  Acknowledgment

We thank the anonymous referees and the action editor for their constructive feedback. This research was supported by the New Zealand MBIE TAIAO Data Science Programme.

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

# A Proofs

4

## A.1 Proof of Theorem 1

*Proof.* We follow the derivations of Theorem 8 in Mansour et al. (2009) and Theorem 1 in Ye & Bors (2022):

$$
\begin{aligned}
\mathcal{L}_{\mathbb{D}}(h, h_y) &\leq \mathcal{L}_{\mathbb{D}}(h, h_{\mathbb{M}}^*) + \mathcal{L}_{\mathbb{D}}(h_{\mathbb{M}}^*, h_{\mathbb{D}}^*) + \mathcal{L}_{\mathbb{D}}(h_{\mathbb{D}}^*, h_y) \\
&\leq \mathcal{L}_{\mathbb{M}}(h, h_{\mathbb{M}}^*) + \mathrm{disc}_L(\mathbb{D}, \mathbb{M}) + \mathcal{L}_{\mathbb{D}}(h_{\mathbb{M}}^*, h_{\mathbb{D}}^*) + \mathcal{L}_{\mathbb{D}}(h_{\mathbb{D}}^*, h_y) \\
&\leq \mathcal{L}_{\hat{\mathbb{M}}}(h, h_{\mathbb{M}}^*) + \widehat{\mathfrak{R}}_{\mathcal{M}}(H) + 3A_0 \sqrt{\frac{\log \frac{2}{\delta}}{2M}} + \mathrm{disc}_L(\mathbb{D}, \mathbb{M}) + \mathcal{L}_{\mathbb{D}}(h_{\mathbb{M}}^*, h_{\mathbb{D}}^*) + \mathcal{L}_{\mathbb{D}}(h_{\mathbb{D}}^*, h_y),
\end{aligned}
\tag{6}
$$

The first inequality is based on the triangle inequality of $\mathcal{L}$. The second inequality is based on the definition of discrepancy distance $\mathrm{disc}_L$. The third inequality is based on the Rademacher Bound (Proposition 2 in Mansour et al. (2009)). □

---

4In the proof, we drop $t$ from the symbols for conciseness.

### A.2 Proof of Proposition 1

*Proof.* Let $\gamma \doteq \frac{N^-}{N}$ and $\alpha \doteq \frac{M_{short}}{M}$. Based on the definition of discrepancy distance, we have:

$$
\begin{aligned}
\mathrm{disc}_L(\mathbb{D},\mathbb{M}) & \\
= \max_{h,h'\in H} &|\gamma\mathcal{L}_{\mathbb{P}^-}(h',h) + (1-\gamma)\mathcal{L}_{\mathbb{P}^+}(h',h) \\
& - \left((1-\alpha)\mathcal{L}_{\mathbb{M}_{long}}(h',h) + \alpha\mathcal{L}_{\mathbb{M}_{short}}(h',h)\right)|.
\end{aligned}
\tag{7}
$$

Since all the samples from the short-term memory come from the current task, $\mathcal{L}_{\mathbb{M}_{short}}(h',h)$ only depends on the current task distribution and is not affected by the size of short-term and long-term memory, i.e. we have $\mathcal{L}_{\mathbb{M}_{short}}(h',h) = \mathcal{L}_{\mathbb{P}^+}(h',h)$. In contrast, the long-term memory is managed by the reservoir sampling method and thus $\mathcal{L}_{\mathbb{M}_{long}}(h',h)$ is affected by the memory allocation factor $\lambda$. Letting $\beta \doteq \frac{N^-}{N-\alpha M}$, we have $\mathcal{L}_{\mathbb{M}_{long}}(h',h) = \beta\mathcal{L}_{\mathbb{P}^-}(h',h) + (1-\beta)\mathcal{L}_{\mathbb{P}^+}(h',h)$. Inserting these two results into Eq 7 gives:

$$
\begin{aligned}
\mathrm{disc}_L(\mathbb{D},\mathbb{M}) & \\
= \max_{h,h'\in H} &|(\gamma - (1-\alpha)\beta))\left(\mathcal{L}_{\mathbb{P}^-}(h',h) - \mathcal{L}_{\mathbb{P}^+}(h',h)\right)| \\
= (\gamma - (1-\alpha)\beta)) &\max_{h,h'\in H} |\left(\mathcal{L}_{\mathbb{P}^-}(h',h) - \mathcal{L}_{\mathbb{P}^+}(h',h)\right)| \\
= \frac{N^-\lambda(N-M)}{N(N+\lambda N - \lambda M)} &\,\mathrm{disc}_L(\mathbb{P}_-,\mathbb{P}_+).
\end{aligned}
\tag{8}
$$

This second equality is based on the fact that $\gamma - (1-\alpha)\beta = \frac{\alpha N^-(N-M)}{N(N-\alpha M)} > 0$ when $N > M$. The third equality is based on the definitions of $\alpha$, $\gamma$, $\beta$, $\lambda$, and $\mathrm{disc}_L$ $\qquad\square$

### A.3 Proof of Corollary 1

$$
\begin{aligned}
\nabla_\lambda \mathrm{disc}_L(\mathbb{D}_t,\mathbb{M}_t) =& \nabla_\lambda \frac{N_t^-\lambda(N_t-M)}{N_t(N_t+\lambda N_t - \lambda M)}\,\mathrm{disc}_L(\mathbb{D}_t^-,\mathbb{D}_t^+) \\
=& \,\mathrm{disc}_L(\mathbb{D}_t^-,\mathbb{D}_t^+)\frac{N_t^-(N_t-M)}{N_t}\nabla_\lambda \frac{\lambda}{N_t+\lambda N_t-\lambda M} \\
=& \,\mathrm{disc}_L(\mathbb{D}_t^-,\mathbb{D}_t^+)\frac{N_t^-(N_t-M)}{N_t}\nabla_\lambda \frac{1}{\frac{N_t}{\lambda}+N_t-M} \\
=& \,\mathrm{disc}_L(\mathbb{D}_t^-,\mathbb{D}_t^+)\frac{N_t^-(N_t-M)}{N_t}\frac{1}{(\frac{N_t}{\lambda}+N_t-M)^2}\frac{N_t}{\lambda^2} \\
=& \frac{N_t^-(N_t-M)}{(N_t+\lambda N_t-\lambda M)^2}\,\mathrm{disc}_L(\mathbb{D}_t^-,\mathbb{D}_t^+)
\end{aligned}
\tag{9}
$$

## B A Review of Memory and Compute Setups in Continual Learning

Online and offline CL are typically studied separately with different memory and compute setups (see Table B). For the compute cost, offline CL papers and surveys typically utilize 50-250 training epochs on standard benchmarks, while online CL by design can only use a single epoch. Moreover, while the gradient update steps in online CL can be increased by using repeated iteration, conducting multiple gradient updates for each incoming batch (Zhang et al., 2022; Soutif-Cormerais et al., 2023), the number of iterations is commonly chosen to be less than 10. For the storage cost, a common practice in CL is comparing different online or offline CL methods based on a fixed budget of exemplars—a subset of past samples. Less discussed is the sample complexity and storage cost introduced by the new task data, which also occupies the constrained space and is used for the training. At each training session, offline CL needs to store the full task datasets for model training, whereas online CL only stores the most recent incoming batches of the new task.

Table 6: The compute and data storage cost of online and offline continual learning on the same CL benchmark Split-CIFAR100.

| | Papers | Data Storage (# samples) | | Compute |
|---|---|---|---|---|
| | | Exemplar | New data | Iteration/Epoch |
| online CL | ER Chaudhry et al. (2019) | 2000, 5000 | 10 | 1 |
| | ER-ACE Caccia et al. (2021) | | | 1 |
| | SCR (Mai et al., 2021) | | | 1 |
| | ER-OBC (Chrysakis & Moens, 2023) | | | 1 |
| | RAR (Zhang et al., 2022) | | | 10 |
| | Survey (Mai et al., 2022) | | | 1,5 |
| | Survey (Soutif-Cormerais et al., 2023) | | | 3 |
| offline CL | ICARL Rebuffi et al. (2017) | 2000,5000 | 2500,5000 | 70 |
| | EEIL Castro et al. (2018) | | | 70 |
| | LUCIR Hou et al. (2019) | | | 160 |
| | DER++ (Buzzega et al., 2020) | | | 50 |
| | MEMO (Zhou et al., 2022) | | | 170 |
| | BIC (Wu et al., 2019) | | | 250 |
| | Survey Masana et al. (2022) | | | 100 |
| online, semi-offline, offline CL | ours | $\frac{1}{1+\lambda}M$ | $\frac{\lambda}{1+\lambda}M$ | 50 |

## C Stability and plasticity analysis

As shown in Section 6, in DER, the new data and exemplar data use different loss functions. In ER, SCR, and iCaRL, the two employ the same type of loss function. The gradients can be rewritten as follows:

$$
\begin{aligned}
\nabla\mathcal{L} &= \mathbb{E}_{\mathcal{M}_{short}}[\nabla\ell] + \mathbb{E}_{\mathcal{M}_{long}}[\nabla\ell] \\
&= \mathbb{E}_{\mathcal{M}_{short}}[\nabla\ell] + p_{new} \times \mathbb{E}_{\mathcal{M}_{long}^{new}}[\nabla\ell] + (1 - p_{new}) \times \mathbb{E}_{\mathcal{M}_{long}^{old}}[\nabla\ell]
\end{aligned}
\tag{10}
$$

Based on $p_{new} = \frac{C - M_s}{N - M_s}$, we have $p_{new} \ll 1$ in our main experiments. In particular, based on the task size and datastream sizes, we have: CIFAR100 $p_{new} \in [0, 0.045]$, Mini-ImageNet $p_{new} \in [0, 0.099]$, CORE-50 $p_{new} \in [0, 0.100]$). Thus, the plasticity changes are dominated by the first term: as the size of the short-term memory decreases, the first term faces greater overfitting on the new task data, harming plasticity. In contrast, DER++ employs different losses for the first and second terms, which leads to different plasticity dynamics as shown in Figure 3.

## D Experiment Setup

**Implementation Details**. Our experiments were conducted on NVIDIA RTX 1080 Ti, 2080 Ti, and A6000 GPUs. In the main experiment, using a batch size of 50, each run for CIFAR100, Mini-ImageNet and CORE50 took roughly 2h, 9h, and 30h respectively. Following Masana et al. (2022), all experiments utilized ResNet-18. We use standard data augmentation (random cropping and flipping), a data stream batch size of 50 for CIFAR100 and Mini-Imagenet, 64 for CORE50. An equal number of exemplars is sampled at each gradient step. All models use vanilla SGD for optimization with a learning rate of 0.1. For iCaRL and SCR, a nearest-class-mean (NCM) classifier is applied as in the original publications. The default iteration and epoch number is 50.

**Hyperparameters**. The hyperparameter settings are summarized in Table 7. The regularization strength in DER++ and the temperature values in SCR follow the original papers.

## E Unified Continual Replay Framework Instantiated with DER++

Table 7: Hyperparameter settings.

| | HYPERPARAMETER |
|---|---|
| ER | LR=0.1 |
| iCaRL | LR=0.1,NCM CLASSIFIER |
| SCR | TEMP =0.07, LR=0.1, NCM CLASSIFIER |
| DER ++ | $\alpha = 0.1$, $\beta = 0.5$,$lr = 0.03$ (CIFAR100) |
| | $\alpha = 0.3$ $\beta = 0.8$,$lr = 0.1$ (MINI-IMAGENET) |
| | $\alpha = 0.1$, $\beta = 1.0$,$lr = 0.1$ (CORE50) |

Algorithm 2 integrates the DER++ method into the framework. Since DER++ uses the past logits output for knowledge distillation, the algorithm stores first the logit outputs from the learning trajectory in the short-term memory (line 17). These logits are then transferred to the long-term memory for use in future knowledge distillation steps (line 5).

---

**Algorithm 2:** Unified continual replay framework instantiated with DER++

```
// Non-stationary data stream:  𝒟ₜ = ∪ₜ𝒳ₜ, where 𝒳ₜ is the incoming batch
// Short-term memory ℳ_short:  storing new data
// Long-term memory ℳ_long:  storing past data
// offline CL: |ℳ_short| is equal to the task size |∪_{Tᵢ≤t<Tᵢ₊₁} 𝒳ₜ|
// online CL: |ℳ_short| is equal to the stream batch size |𝒳ₜ|
// π is a memory management policy
// z is logits output and scalars α,β are regularization hyperparameters of DER++
```

1 **function** ContinualLearning($\mathcal{X}_t, \theta, \mathcal{M}_{short}, \mathcal{M}_{long}$,)
2 $\quad \mathcal{M}_{short} \leftarrow \mathcal{M}_{short} \cup \mathcal{X}_t$ // Update short-term memory
3 $\quad$ **if** $\mathcal{M}_{short}$ **is** full **then**
4 $\quad\quad \theta, \mathcal{M}_{short} \leftarrow$ ModelTraining($\theta, \mathcal{M}_{short}, \mathcal{M}_{long}$)
5 $\quad\quad \mathcal{M}_{long} \subset_\pi \mathcal{M}_{short} \cup \mathcal{M}_{long}$ // Update long-term memory
6 $\quad\quad \mathcal{M}_{short} \leftarrow \emptyset$
7 $\quad$ **return** $\theta, \mathcal{M}_{short}, \mathcal{M}_{long}$

8 **function** ModelTraining($\theta, \mathcal{M}_{short}, \mathcal{M}_{long}$)
9 $\quad$ **for** K epochs **do**
10 $\quad\quad$ **for** $\mathcal{B}_{short} = \{x, y\}$ **in** $\mathcal{M}_{short}$ **do**
11 $\quad\quad\quad$ sample $\mathcal{B}'_{long} = \{x', y', z'_{prev}\}$ from $\mathcal{M}_{long}$,
12 $\quad\quad\quad$ sample $\mathcal{B}''_{long} = \{x'', y'', z''_{prev}\}$ from $\mathcal{M}_{long}$,
13 $\quad\quad\quad x, x', x'' \leftarrow aug(x'), aug(x'), aug(x'')$
14 $\quad\quad\quad z = h_\theta(x)$
15 $\quad\quad\quad L_{DER++} = \ell_{ce}(x, y; \theta) + \alpha \left\| z'_{prev} - h_\theta(x') \right\|_2^2 + \beta \ell_{ce}(x'', y''; \theta)$
16 $\quad\quad\quad \theta \leftarrow \theta - \eta \nabla L_{DER++}$
17 $\quad\quad\quad$ update $\mathcal{M}_{short}$ with logits information $\mathcal{B}_{short} = \{x, y, z\}$
18 $\quad$ **return** $\theta, \mathcal{M}_{short}$

---

## F    Additional Experimental Results

The main paper shows the stability and plasticity analysis with Mini-ImageNet. Here, Fig 4 shows the results with CIFAR100 with different task numbers (20 tasks vs. 50 tasks) and across architectures (ResNet-18 and ResNet-34). These new results show behavior consistent with our previous findings. For ER, SCR, and ICARL, a smaller memory consumption ratio (MCR) leads to increased stability but decreased plasticity.

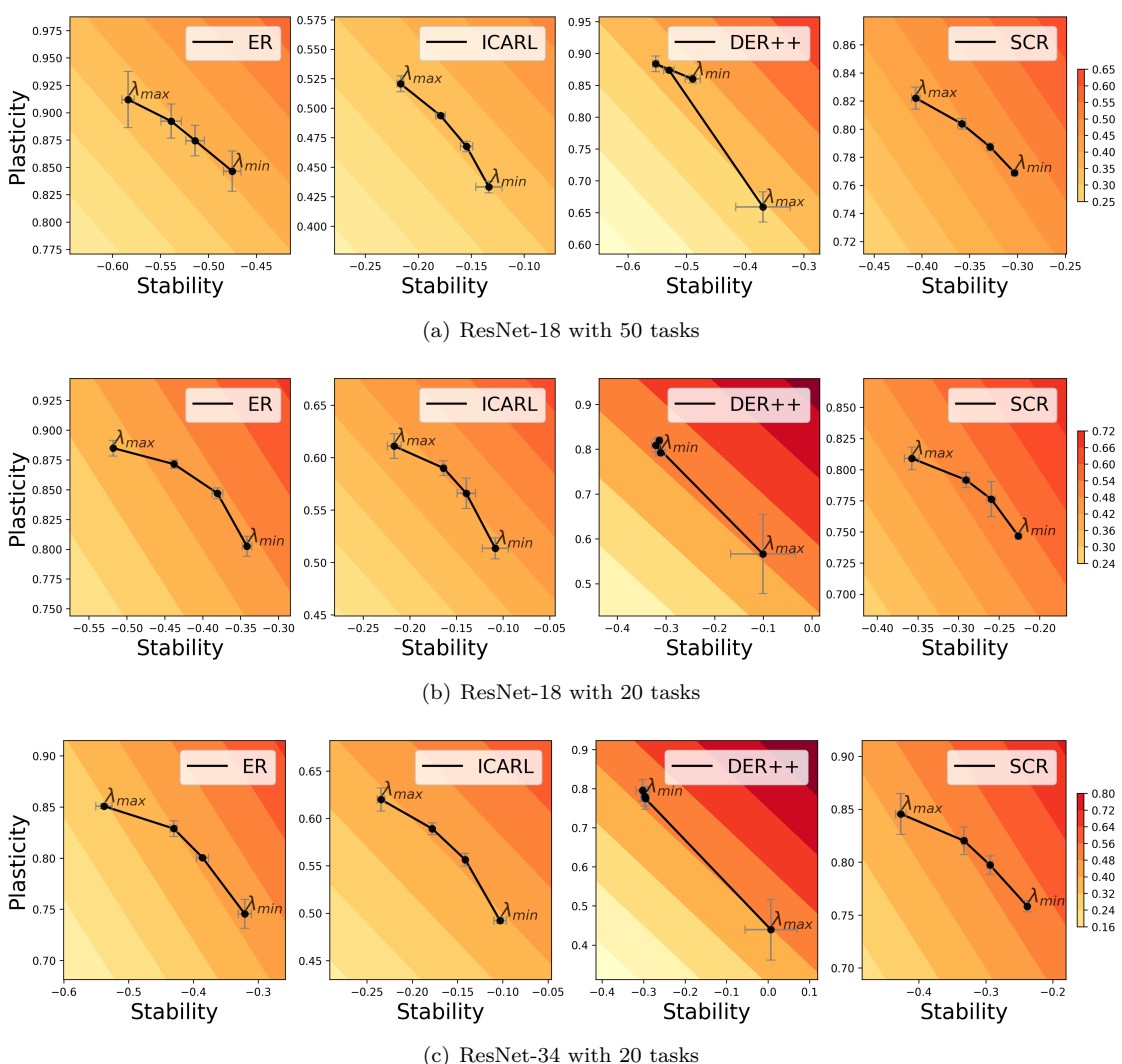

(a) ResNet-18 with 50 tasks

(b) ResNet-18 with 20 tasks

(c) ResNet-34 with 20 tasks

Figure 4: Stability and plasticity analysis with different model architectures (ResNet18 and ResNet34) and different task numbers (20 and 50 tasks) in CIFAR100.

For DER++, reducing the MCR results in different stability-plasticity behaviors, such as improved plasticity or improved stability at certain stages.

