# OpenReview forum: "Rethinking Memory in Continual Learning: Beyond a Monolithic Store of the Past"
_TMLR — Accepted by TMLR_

### Review · Reviewer_LdYh · 2025-08-19

**Summary Of Contributions:**

This work introduces a novel concept of dual memory for online and offline continual learning (CL) scenarios. Traditionally, memory in replay-based CL methodologies is considered as a single unit consisting of past representative examples. The authors propose looking at the interplay between past and current examples to strike a balance between plasticity and stability. They introduce a new metric called memory capacity ratio (MCR) to describe the relative number of past and new examples to store. They present findings on how MCR impacts the performance of online and offline CL methods.

Strengths:
1) Introduction of the concept of structured memory is conceptually sound and applicable for both online and offline settings.
2) Presenting a quantifiable metric to study the effect of short term and long term memory on stability and plasticity enables important insights, like using a smaller MCR yields higher generalization. Such findings are particularly useful for practical applications.
3) The experiments are well thought out by inclusion of different control variables for a fair comparison of results.

Weaknesses:
1) A smaller MCR implies fewer new examples for the model to learn new patterns. This can translate to slower adaptation to new data or data distributions. This can be tested, for example,  with fast switches between tasks or simulating domain shifts between tasks. The authors have not presented results under such conditions.
2) Like the authors have mentioned under “Limitations”, there are no insights provided on sample selection strategies. With a lower MCR or short term memory, sample selection from current or recent tasks becomes critical for model updates. Incorporating a random reservoir sampling for exemplar selection might be the reason for differing stability/plasticity trends with different algorithms.
3) While a controlled experimental setup enables better comparison between methods, fixed hyperparameters across datasets can lead to suboptimal performance and incorrect interpretation of results. It would be helpful to include more details on how the hyperparameters were selected and specify any assumptions made about them when interpreting the results.
4) The paper explores and presents results on a range of MCR values. It may have been more valuable to provide an objective or method to find an optimal MCR.

**Audience:**

Yes

**Audience Explanation:**

Applications with memory constraints like edge and IoT devices would benefit from advances in CL. Another important use case for this work is in domains like healthcare where privacy is critical and long term storage is not possible.

**Broader Impact Concerns:**

None.

**Claims And Evidence:**

Yes

**Claims Explanation:**

Aside from the weaknesses mentioned which may complicate interpretation, the claims are supported by clear evidence.

**Requested Changes:**

1) My main areas of concern is using fixed hyperparameters across datasets and algorithms and how it could be suboptimal for different MCR settings.
2) One of the major findings is that a lower MCR can yield comparable or better performance compared to a higher MCR. This claim can be made more convincing by designing more extensive experiments like mentioned earlier.
3) Minor: Figure 3 may be simplified. The caption can also be improved.

---

> ### Author Response · Authors · 2025-10-17
> **Reply to Reviewer LdYh: new results and changes in the revised manuscript**
>
> We thank **Reviewer LdYh** for the thorough evaluation of our work and for the insightful suggestions and feedback. Below, we provide point-by-point responses, including new results, clarifications, and changes. The revised manuscript is uploaded with all changes highlighted in blue.
>
> ---
>
> **1. Sample Selection Strategies**
>
> Thank you for the feedback. To examine the effect of sample selection strategies on our findings, we added additional results using two advanced selection methods, MIR and ASER, and reported them in **Section 5 (Table 5)**.
>
> Unlike reservoir sampling, these methods select samples based on interference scores rather than uniform probability. Our findings remain consistent with both MIR and ASER: a smaller MCR leads to notable performance improvements.
>
> **Rebuttal Table 1. Results with different sample selection methods.**
> | Method | MCR | Mini-ImageNet | CIFAR-100 |
> |-|-|-|-|
> | **MIR** | MCR_min | **43.4 ± 0.3** | **43.4 ± 3.1** |
> |   | MCR_mid | 40.6 ± 0.5 | 41.4 ± 1.4 |
> |   | MCR_max | 30.8 ± 0.6 | 33.4 ± 2.8 |
> | **ASER** | MCR_min | **40.6 ± 0.8** | **42.8 ± 1.7** |
> | | MCR_mid | 38.2 ± 0.6 | 41.0 ± 2.2 |
> |    | MCR_max | 31.1 ± 1.0 | 33.8 ± 1.2 |
>
> We also added a discussion in **Section 7** to highlight a future research direction as developing sample selection with time-varying MCR.
>
> ---
>
> **2. Adaptation to new tasks ( fast-switch tasks)**
>
> Thank you for the insightful suggestion. We performed additional experiments on fast-switch tasks to investigate adaptation ability. While the main paper reports results on CIFAR-100 split into 20 tasks, we constructed a fast-switch dataset by splitting CIFAR-100 into 50 tasks, each containing 2 classes.
>
> Adaptation ability to new tasks is quantified by plasticity (as defined in Section 6). Results are reported in **Figure 4(a) in Appendix F**.
>
> Consistent with our earlier findings, we observe that a smaller MCR does _not_ always slow adaptation—it depends on the replay strategy. For ER, ICARL, and SCR, reducing MCR usually decreases plasticity, while for DER++, a smaller MCR can improve plasticity, at the cost of stability.
>
> ---
>
> **3. Hyperparameter setup**
>
> To address the concern regarding hyperparameter effects on MCR, we examined a key parameter, the learning rate, under different MCR values. Our main experiments used a learning rate of 0.1. Inspired by the reviewer’s feedback, we added results for learning rates of 0.01 and 0.3. As shown below, the advantage of a small MCR holds consistently across learning rates.
>
>
> **Rebuttal Table 2. Results with different learning rates.**
>  | lr   | incoming_buffer_size | Test Accuracy    |
> |---|---|--|
> |CIFAR100|
>  | 0.01 | 50 | **40.9**    |
>  |  | 1250    | 39.1   |
> |  | 2500     | 31.5 |
> | 0.3  | 50   | **40.6**    |
> |   | 1250     | 38.4     |
> |   | 2500    | 31.5      |
> |Mini-ImageNet|
> | 0.01 | 50  | **44.5**      |
> |  | 1250  | 41.3      |
> |  | 2500   | 31.7      |
> | 0.3  | 50   | **36.1**   |
> |   | 1250   | 33.9   |
> |   | 2500  | 25.9   |
>
> To further ensure robustness, we conducted additional experiments following the online continual learning (OCL) hyperparameter tuning protocol [1,2].
> Unlike traditional tuning, which uses held-out validation data from all tasks (violating the assumption of no access to future tasks), OCL tuning uses a separate validation stream, e.g., the first few tasks. We used the first 3 tasks of CIFAR-100 and Mini-ImageNet for validation and the _remaining tasks_ for training/testing, performing a grid search over learning rates [0.001, 0.01,0.1]. Results consistently show that MCR_min values yield comparable or better performance than MCR_max.
>
> **Rebuttal Table 3. Results with online hyperparameter tuning.**
> |  | ER    | ICARL | DER++ | SCR   |
> |-|-|-|-|-|
> | CIFAR100 |
> | MCR_min   | **48.4** | **44.4** | **50.1** | **55.1** |
> | MCR_max  | 39.6 | 43.1 | 45.2 | 47.6 |
> | Mini-ImageNet |
> | MCR_min  | **52.6** |	**50.3**|	**54.0**|	**56.6**|
> | MCR_max  | 40.5	|49.7| 50.2 |51.9|
>
> Regarding other hyperparameters, such as regularization strength, we follow the choice in [3], which uses dataset-specific DER++ regularization settings (as shown in Table 7 in the Appendix).
>
> ---
>
> **4. Improve Fig 3**
>
> We have **updated Fig 3** by including results over different model initialization and task splits. We also revised the caption for clarity. In addition, we added another **Fig 4 (in Appendix F)** to validate the findings of  Fig 3 with other datasets and architectures.
>
> *[1] Mai, Zheda, et al. Online continual learning in image classification: An empirical survey. Neurocomputing 469 (2022): 28-51.*
>
> *[2] A. Chaudhry, et al. Efficient lifelong learning with a-GEM, ICLR, 2019.*
>
> *[3] Boschini, Matteo, et al. "Class-incremental continual learning into the extended der-verse." IEEE transactions on pattern analysis and machine intelligence 45.5 (2022): 5497-5512.*

---

### Review · Reviewer_wpCo · 2025-10-02

**Summary Of Contributions:**

A very clearly written paper, even for a non-theory person.
1. Proposed a continual learning method which uses short and long term memory.
2. Deeply explores the role of short divided by long term memory (called MCR)
3. Touches upon plasticity, stability, and generalization aspects
4. Comprehensive empirical justification with studying one outlier replay algorithm

**Audience:**

Yes

**Audience Explanation:**

The paper may invigorate some people's thinking about continual learning online-offline hybrid systems. The theoretical work is not much in this paper and also makes a lot of assumptions. So, it may attract people to make more complex extensions to this.

**Broader Impact Concerns:**

Definitely opens up conversation in continual learning which combines online and offline learning due to their unified framework.
While the method proposed is quite simple, I can see much more sophisticated extensions to this. This paper, per my guess, will mostly serve as motivation for other people to improve upon (non-trivial achievement) but the paper itself is simple technical contribution.

**Claims And Evidence:**

Yes

**Claims Explanation:**

1. There is clear algorithm description
2. There is some theoretical analysis to show their proposal reduces generalization bound
3. Ample empirical proof and individual replay algorithm study provided

**Requested Changes:**

1. Clarify. In class incremental setting, do model output neurons whose class labels have not "arrived yet", do they remain waiting? In simpler words, in CL, does model architecture remain unchanged?
2. Clarify. Why did you choose a very simple STM and LTM storage scheme (greedy and reservoir sampling resp.)? I was expecting more. Atleast explain why this simple choice suffices.
3. In theorem 1, pi i.e. sampling algorithm does not play a role? Clarify.
4. At the end of theorem 1, you did not give final comments. So low MCR leads to low DISC_L - thereby reducing generalizartion error bound. But this was kind of expected result (not discrediting the work BTW). I feel there is some baseline missing so say you are theoretically outperforming it. Or maybe baseline is high MCR itself. Clarify
5. Try to give some citation for true labeling function deterministic assumption you made.
6. Try to improve Fig 3 placement. You start talkign about soon enough but figure is far away.
7. Try to Comment and analyze why MCR can't be variable with time. Readers will find value in your paper if you leave some hope for extension towards the end. Your algorithm is a bit simple and you did not compare reservoir sampling with anything else so I am thinking on your behalf to make paper stronger. Think about it.
8. Please bold numbers in your tables to show which numbers are best. It will improve readability. Tables looks bland right now and there are some inconsistent formattings.
9. You did not define DI, TI, "pretrained model" settings in Table 4. I could not fully understand the comparisons to CI.

---

> ### Author Response · Authors · 2025-10-17
> **Reply to Reviewer wpCo: clarifications, new results and changes in the paper**
>
> We thank Reviewer **wpCo** for the thorough evaluation of our work and for providing insightful suggestions and feedback. Below, we respond point by point, providing new results and clarifications. The revised version is uploaded with all changes highlighted in blue.
>
> ---
>
>  **1. Clarification on the class-incremental setting**
>
> When a new task arrives, the model backbone architecture remains unchanged, but new output neurons are added to the classification head. For example, in the Split-CIFAR100 benchmark with 10 tasks, the classifier head initially contains 10 classes. When the second task is introduced, the output head expands to 20 classes, and so on.
>
> ---
>
> **2. STM and LTM storage scheme**
>
> Thank you for this detailed feedback.  For LTM, we focus on reservoir sampling as it is a widely used and theoretically analyzable method. For STM, we employ a greedy mechanism to maintain consistency with existing online and offline CL practices and to ensure that all samples in the data stream are seen by the model at least once.
>
> We acknowledge that developing advanced strategies for selecting LTM samples is an active area of research. To address this, we added additional results using two advanced selection methods—**MIR** and **ASER**—in **Section 5 (Table 5)**. Unlike reservoir sampling, these methods select samples based on interference scores rather than equal probability. Our findings are consistent with MIR and ASER: a smaller MCR leads to notable performance gains.
>
> | Method | MCR | Mini-ImageNet | CIFAR-100 |
> |--|-----|--|----|
> | **MIR** | MCR_min | **43.4 ± 0.3** | **43.4 ± 3.1** |
> |         | MCR_mid | 40.6 ± 0.5 | 41.4 ± 1.4 |
> |         | MCR_max | 30.8 ± 0.6 | 33.4 ± 2.8 |
> | **ASER** | MCR_min | **40.6 ± 0.8** | **42.8 ± 1.7** |
> |          | MCR_mid | 38.2 ± 0.6 | 41.0 ± 2.2 |
> |          | MCR_max | 31.1 ± 1.0 | 33.8 ± 1.2 |
>
> ---
>
> **3. The role of π in Theorem 1**
>
> Thank you for the insightful question. We have expanded the discussion in **Section 4** to clarify the role of the sample selection policy **π**.
>
> The policy **π** governs the relationship between the distribution of stored samples (**M**) and the data stream (**D**). **Theorem 1** holds regardless of the specific relationship between **M** and **D**, since its proof does not assume similarity between them. However, the degree of alignment between **M** and **D** influences the tightness of the bound through the term L_D(h*_M, h*_D) on the right-hand side of Theorem 1.
>
> This term quantifies the discrepancy between the optimal hypotheses for **M** and **D** within the hypothesis space. Since both **M** and **D** include samples from old and new task domains, and assuming a sufficiently expressive model, this term is expected to be small—thus the bound is mainly governed by disc_L. Conversely, if **M** and **D** differ substantially, this term grows large, loosening the bound. We added this clarification in **Section 4** to emphasize the theoretical implications of **π**.
>
> ---
>
>  **4. Remarks on Theorem 1**
>
> Thank you for this question. We have added remarks in the paper to clarify the implications of **Theorem 1**.
> The theorem highlights that a key factor affecting generalization performance is the discrepancy disc_L(M,D), which measures the difference between the **stored data** and the data stream. While much prior work has focused on constructing representative exemplars (the **long-term memory**), the role of the **incoming buffer** (short-term memory) is often overlooked. Theorem 1 reveals that it is the **joint distribution** of long-term and short-term memories—rather than the long-term memory alone—that determines the generalization bound.
>
> ---
>
>  **5. Citations for deterministic label assumption**
>
> We have added two references.
>
> ---
>
> **6. Figure 3 placement**
>
> We have adjusted the placement of **Figure 3** and corrected a typo in **Section 5.2**—the text originally referred to *Figure 3* but should have referenced *Figure 1*.
>
> ---
>
>  **7. Time-varying MCR**
>
> Thank you for this excellent suggestion. We added a discussion on time-varying MCR as a promising direction for future work in **Section 7**.  Designing an adaptive MCR schedule, either across tasks or within each task, could dynamically balance stability and plasticity to further improve performance. However, naïve selection methods such as reservoir sampling may fail under time-varying MCR, since changes in the size of long-term memory alter the storage probability for each task. Developing a suitable sample selection mechanism for this setting is an interesting avenue for future research.
>
> ---
>
>  **8. Table formatting**
>
> We have improved table formatting by adding gain values and bolding key numbers to enhance readability.
>
> ---
>
>  **9. Clarification of Table 4 setup**
>
> The **Table 4 caption** has been revised to clearly describe the *Domain-Incremental* and *Task-Incremental* settings, as well as the pretrained model configurations used.
>
> ---

---

### Review · Reviewer_GRfr · 2025-10-02

**Summary Of Contributions:**

The paper proposes a new framework under which several CL methods can be placed; this framework consists of a pool of long-term memory and a pool of short-term memory. In this framework, the authors define MCR as the ratio between short term and long term memory sizes. The authors then theoretically show that a smaller MCR yields a better generalization bound. Intuitively, this is because long-term memory is more representative of the true current data distribution than short-term memory. Empirically, the authors find that for common CL methods, MCR is indeed predictive of performance: lower MCR performs better.

**Audience:**

Yes

**Audience Explanation:**

Yes: the paper presents a novel framework to analyse generalization in CL which is interesting in itself given the difficulty of theoretical analysis in this field. Moreover, the paper makes the claim that small MCR (as in online RL) can actually outperform large MCR algorithms (as in offline RL) which may be counter-intuitive. Regardless of whether these claims hold more generally beyond the settings which the authors consider, the paper would definitely be of interest to those aiming to understand CL from a theoretical perspective.

**Broader Impact Concerns:**

No broader impact concerns.

**Claims And Evidence:**

Yes

**Claims Explanation:**

The theoretical analysis is solid, adapting existing results in the literature to the proposed framework. The experiments are also sufficient: the paper considers a number of standard CL techniques over multiple datasets. The stability analysis is also interesting although perhaps would benefit from further study (in another paper).

**Requested Changes:**

**Would strengthen**
- Some may find it unclear how many different CL algorithms can fit into the structure of Algorithm 1; I recommend explaining how common algorithms like DER++ or ICARL are special cases of the algorithm
- pi is not defined in line 5 of Algorithm 1
- It's unclear how robust the results in section 6 are; I would recommend including results from multiple datasets & architectures to ensure the results are consistent across settings
- Error bars/circles in Figure 3
- Enlarge Figure 2 a little bit

---

> ### Author Response · Authors · 2025-10-17
> **Reply to Reviewer GRfr: additional results and changes of manuscript**
>
> We thank Reviewer **GRfr** for the thorough evaluation of our work and for providing insightful suggestions and feedback to help improve the paper. Below, we respond point by point, providing new experimental results, clarifications, and indicating the corresponding changes in the revised manuscript. The revised version is uploaded with all changes highlighted in blue.
>
> ---
>
> **1.  Integrate Algorithm 1 with other replay methods**
>
> Thank you for the suggestion. We have added the algorithmic framework incorporating **DER++** in **Appendix E**.  Since DER++ uses the past logits output for knowledge distillation, the algorithm stores first the logit outputs from the learning trajectory in the short-term memory (line 17). These logits are then transferred to the long-term memory for use in future knowledge distillation steps (line 5).
>
> ---
>
> **2. Missing definition of π in Algorithm 1**
>
> We appreciate your careful observation. The definition of **π** (the memory management policy) has been added to **Algorithm 1**. The policy **π** determines how samples are selected from the short-term memory (**M_short**) for storage in the long-term memory (**M_long**).
>
> ---
>
> **3. Robustness of results in Section 6**
>
> Thank you for the detailed feedback. We have included additional **stability–plasticity analyses** across different datasets and architectures in **Appendix F**.
> While the main paper presents results on **Mini-ImageNet**, **Figure 4** now extends this evaluation to **CIFAR-100**, considering varying task numbers (20 vs. 50) and architectures (**ResNet-18** and **ResNet-34**).
> The results are consistent with our main findings:
> 1. For ER, SCR, and ICARL, a smaller MCR mostly improves stability but reduces plasticity.
> 2. For DER++, a smaller MCR produces more varied behavior, occasionally improving plasticity at the expense of stability.
>
> ---
>
> **4. Error bars in Fig3**
>
> We have added results with different initialization and task splits in Fig 3 to provide a better representation of variability across runs.
>
> ---
>
> **5. Enlarging Figure 2**
>
> We have enlarged **Figure 2** to improve readability and clarity.

---

### Decision · Action_Editor_HibM · 2025-11-26

**Recommendation:** Accept as is

**Audience:**

Yes

**Audience Explanation:**

A novel framework is introduced which is of interest for a broader audience interested in continual learning.

**Claims And Evidence:**

Yes

**Claims Explanation:**

The manuscript contains solid theoretical analysis backed up experimental validation using several data sets.